# Mutation Profile Variability in the Primary Tumor and Multiple Pulmonary Metastases of Clear Cell Renal Cell Carcinoma. A Review of the Literature and Analysis of Four Metastatic Cases

**DOI:** 10.3390/cancers13235906

**Published:** 2021-11-24

**Authors:** Kristyna Prochazkova, Nikola Ptakova, Reza Alaghehbandan, Sean R. Williamson, Tomáš Vaněček, Josef Vodicka, Vladislav Treska, Joanna Rogala, Kristyna Pivovarcikova, Kvetoslava Michalova, Maryna Slisarenko, Milan Hora, Michal Michal, Ondrej Hes

**Affiliations:** 1Department of Surgery, Faculty of Medicine in Pilsen and University Hospital Pilsen, Charles University, 304 60 Pilsen, Czech Republic; Prochazkovak@fnplzen.cz (K.P.); vodicka@fnplzen.cz (J.V.); treska@fnplzen.cz (V.T.); 2Second Faculty of Medicine, Charles University, 150 06 Prague, Czech Republic; ptakova@biopticka.cz; 3Department of Pathology, University of British Columbia, Vancouver, BC 2329, Canada; Reza.Alaghehbandan@fraserhealth.ca; 4Robert J. Tomsich Pathology and Laboratory Medicine Institute and Glickman Urological Institute, Cleveland Clinic, Cleveland, OH 44195, USA; williamson.sean@outlook.com; 5Department of Pathology, Faculty of Medicine in Pilsen and University Hospital Pilsen, Charles University, 305 99 Pilsen, Czech Republic; vanecek@bioptica.cz (T.V.); superrrogalik7@gmail.com (J.R.); Pivovarcikova@fnplzen.cz (K.P.); Kvetoslava.Michalova@biopticka.cz (K.M.); MarynaSlisarenko@gmail.com (M.S.); Michal@biopticka.cz (M.M.); 6Department of Urology, Faculty of Medicine in Pilsen and University Hospital Pilsen, Charles University, 305 99 Pilsen, Czech Republic; horam@biopticka.cz

**Keywords:** clear cell renal cell carcinoma, intra-tumoral heterogeneity, inter-tumoral heterogeneity, inter-metastatic heterogeneity

## Abstract

**Simple Summary:**

Clear cell renal cell carcinoma (CCRCC) is well known for intra-tumoral heterogeneity. However, there are limited data focusing on the inter-tumoral and inter-metastatic heterogeneity of CCRCC. In one study, primary and metastatic tumors were classified as clear cell type A or B subtypes, using nanostring expression technology. It was found that primary and metastatic tumors of CCRCC differed in nearly one half of patients. Approximately one quarter of metastatic tumors display inter-metastatic heterogeneity. Another study, using an immunohistochemical assay, found inter-metastatic tumor heterogeneity of BAP1 in only 1 of 32 patients (3%). Comparing gene expression across patient-matched primary-metastatic tumor pairs, 98% had concordant BAP1 status. We aimed to review published data and to examine mutation profile variability in primary and multiple pulmonary metastases (PMs) in our cohort of four patients with metastatic CCRCC.

**Abstract:**

(1) Background: There are limited data concerning inter-tumoral and inter-metastatic heterogeneity in clear cell renal cell carcinoma (CCRCC). The aim of our study was to review published data and to examine mutation profile variability in primary and multiple pulmonary metastases (PMs) in our cohort of four patients with metastatic CCRCC. (2) Methods: Four patients were enrolled in this study. The clinical characteristics, types of surgeries, histopathologic results, immunohistochemical and genetic evaluations of corresponding primary tumor and PMs, and follow-up data were recorded. (3) Results: In our series, the most commonly mutated genes were those in the canonically dysregulated VHL pathway, which were detected in both primary tumors and corresponding metastasis. There were genetic profile differences between primary and metastatic tumors, as well as among particular metastases in one patient. (4) Conclusions: CCRCC shows heterogeneity between the primary tumor and its metastasis. Such mutational changes may be responsible for suboptimal treatment outcomes in targeted therapy settings.

## 1. Introduction

Clear cell renal cell carcinoma (CCRCC) is the most common renal carcinoma, accounting for more than 70% of adult renal cancer [1,2]. Nonsurgical therapy for metastatic RCC (mRCC) has limited efficacy, with a median overall survival (OS) of 26.4–32.0 months [2]. The lung is one of the most affected metastatic sites in patients with CCRCC. If clinically feasible, metastasectomy is preferable for metastatic disease [3]. The 5 year survival rates after a complete pulmonary metastasectomy range from 36 to 83% [4].

CCRCC is well known for intra-tumoral heterogeneity [2,5,6,7,8,9,10] and morphologic, immunohistochemical and genetic differences also exist between the primary tumor and its metastases (inter-tumoral heterogeneity) [11,12,13,14]. Furthermore, heterogeneity among multiple metastases in a single patient (inter-metastatic heterogeneity) has been reported [11,14].

*VHL*, *BAP1*, *PBRM1,* and *SETD2* are the most frequently mutated genes, all located on chromosome 3p. Chromosome arm 3p loss is a common event in primary CCRCC, and in difficult diagnostic pathology cases, molecular evaluation can be used to support a diagnosis of CCRCC, such as chromosome 3p loss (FISH, cytogenetics, or copy number analysis) or VHL mutational analysis. However, 3p loss may not be entirely specific for clear cell RCC in all contexts [15]. For example, chromosome 3p loss has been recognized in subsets of papillary RCC, unclassified RCC, and RCC with the amplification of the 6p21/*TFEB* gene region, including in tumors with non-clear cell morphology and without *VHL* alterations [16,17,18]. Although the majority of CCRCCs show mutation in the *VHL* gene, LOH3p, or the hypermethylation status of *VHL* gene, 25–30% of CCRCCs show other molecular genetic changes [2]. The molecular study of the Cancer Genome Atlas Research Network identified 19 significantly mutated genes, with alterations of *VHL*, *PBRM1*, *SETD2*, *KDMC*, *PTEN*, *BAP1*, *MTOR* and *TP53*, being the eight most frequent [2,19].

CCRCC is ideal for studying intra-tumoral heterogeneity, since adjuvant therapy is not standard practice [3]. Therefore, the effect of therapy on the development of resistance or tumor changes can be excluded. The aim of this review was to summarize the current knowledge on intra-tumoral, inter-tumoral, and inter-metastatic heterogeneity in CCRCC at the morphologic, immunohistochemical, and molecular-genetic levels.

### 1.1. Morphology and Immunohistochemistry

#### 1.1.1. Intra-Tumoral Heterogeneity

López et al. [5] drew attention to the problem of tumor sampling, particularly in CCRCC where some large tumors may display areas with different colors and/or textures on gross sections. It is worth noting that even neoplastic cell populations in CCRCC, which may seem homogenous microscopically, indeed may be very heterogeneous at the molecular level with different mutation profiles in different parts of the tumor [6]. In routine clinical practice, more than 95% of the tissue of a given 10 cm tumor is not analyzed, when following typical sampling protocols (i.e., one block per 1–2 cm of the tumor). In these cases, the histo-molecular data that might be derived from non-sampled areas of the tumor are lost. Therefore, some authors suggest that a multisite tumor sampling approach would be more informative than routine sampling [6,7].

CCRCC is typically immunoreactive for PAX8, PAX2, pankeratin (AE1–AE3), CAM5.2, and epithelial membrane antigens. Carbonic anhydrase 9 (CA9) is positive in a diffuse membranous pattern in 75–100% of CCRCC; however, high-grade tumors may exhibit a reduced immunohistochemical expression [2]. According to the latest edition of the WHO classification of genitourinary tumors, keratin 7 positivity in CCRCC is only seen in isolated cells, in rare high-grade tumors, and is often used to distinguish CCRCC from chromophobe RCC [2]. However, in a recent study by Gonzalez et al. examining keratin 7 reactivity in a spectrum of 75 CCRCC tumors, it was shown that low-grade CCRCCs were more frequently positive than high-grade tumors [8].

#### 1.1.2. Inter-Tumoral Heterogeneity

Eckel-Passow and colleagues analyzed the immunohistochemical expression of BAP1 and PBRM1 in primary and metastatic tumors from 97 patients. In their cohort, 20% of primary tumors showed the loss of BAP1 staining and 57% showed the loss of PBRM1. They demonstrated subtle molecular heterogeneity in the metastatic tumors with similar morphology. Comparing expression across patient-matched primary-metastatic tumor pairs, the authors reported that 98% had concordant BAP1 status (90% PBRM1). Only two patients demonstrated discordant BAP1 immunohistochemical expression, with the loss of BAP1 during the progression to metastatic disease [11].

#### 1.1.3. Inter-Metastatic Heterogeneity

Eckel-Passow et al. [11] also determined the inter-metastatic tumor heterogeneity of BAP1 using immunohistochemical examination. However, they found heterogeneity of BAP1 in only 1 patient in a cohort of 32 patients (3%). The primary tumor for this patient was BAP1 positive, whereas the first bone metastasis was IHC negative, and the second bone metastasis) was IHC positive. In this study, the authors also examined intra-metastatic tumor heterogeneity, and found a 100% concordance in BAP1 between 12 patients. The limitation of this study was that the expression was determined using an immunohistochemical assay only, with no further molecular genetic validation.

### 1.2. Molecular Genetic Analysis

#### 1.2.1. Intra-Tumoral Heterogeneity

Gerlinger et al. analyzed material from four tumors (core biopsy) in four patients with metastatic CCRCC. They demonstrated intra-tumoral heterogeneity for a mutation within an auto-inhibitory domain of the mTOR kinase. Mutational intratumoral heterogeneity was found for multiple tumor suppressor genes resulting in a loss of function. Multiple distinct mutations of *SETD2*, *PTEN,* and *KDM5C* genes were found within a single tumor [9].

In their subsequent study, the authors showed that ultra-deep sequencing identified intra-tumoral heterogeneity in all cases. Using multiregional exome sequencing, the authors reported the following as the most prevalent mutations: *PBRM1* 60%, *SETD2* 30%, *BAP1* 40%, *KDM5C* 10%, *TP53* 40%, *ATM* 10%, *ARID1A* 10%, *PTEN* 20%, *M**TOR* 10%, *PIK3CA* 20%, and *TSC2* 10%. The combined prevalence of the indicated PI3K-mTOR pathway genes (*PTEN*, *PIK3CA*, *TSC2*, *MTOR*) was up to 60% [10].

#### 1.2.2. Inter-Tumoral Heterogeneity

According to Serie et al. [14], heterogeneity between primary and distant simultaneous metastases affects half of the patients with metastatic CCRCC. The authors analyzed primary CCRCC and their metastases using nanostring technology. Nanostring assays were successful in 91 primary tumors and 123 metastases from different organs, most frequently from the lung. ClearCode 34 genes were also analyzed for all tumors. They divided primary and secondary tumors into so-called ccA and ccB subtypes, based on the proposed stratification by Brooks et al. [12]. They further compared ccA/ccB subtypes across patient-matched primary and metastatic CCRCC tumors and documented discordance in 43% of patients.

#### 1.2.3. Inter-Metastatic Heterogeneity

Serie et al. [14] also evaluated inter-metastatic tumor heterogeneity. Thirty patients in their cohort had more than one metastatic tumor. Seven of the 30 (23%) had metastatic tumors with discordant ccA/ccB subtypes.

## 2. Materials and Methods

Pulmonary metastasectomy for metastatic CCRCC (single or multiple metastases) was performed in 35 patients (without evidence of local residual disease, recurrence, or any disease other than pulmonary metastases) in a single academic institution (Department of Surgery, University Hospital in Pilsen) from January 2001 to January 2019. From this cohort, 13 patients had undergone multifocal surgical treatments for their pulmonary metastases of CCRCC. Four patients were excluded from our study since the primary tumor was not available. Five patients were later excluded from the study because of low DNA quality. Finally, four cases were selected and enrolled into the study.

The following clinical and pathologic characteristics were obtained: gender, age at diagnosis, tumor size, pathologic stage [20], histologic grade (ISUP/WHO) [2], progression-free interval (PFI is defined as the time period between curative primary kidney surgery and the first detection of metastatic disease), pulmonary metastases details (site, size of the largest metastasis, synchronous or metachronous, number, and laterality), the type of pulmonary surgery, histopathology results, the type of adjuvant therapies, and follow-up data.

The primary tumor was diagnosed based on morphology and the immunohistochemical (IHC) profile. The tissues were processed as published previously [21]. The following primary antibodies were used: keratin 7 (OV-TL12/30, monoclonal, DakoCytomation, 1:200), vimentin (D9, monoclonal, NeoMarkers, Westinghouse, CA, USA, 1:1000), carbonic anhydrase 9 (rhCA9, monoclonal, R&D Systems, Abingdon, GB, USA, 1:100), PD-L1 (22C3, monoclonal, Cell Signaling, Danvers, MA, USA, 1:25), and Ki67 (MIB1, monoclonal, Dako, Glostrup, Denmark, 1:1000). The primary antibodies were visualized using a supersensitive streptavidin–biotin–peroxidase complex (BioGenex, Fremont, CA, USA). Internal biotin was blocked by the standard protocol used by the Ventana BenchMark XT automated stainer (hydrogen peroxide-based). Appropriate positive and negative controls were applied. The immunohistochemical evaluation was based on the staining percentage of cells: focal positive < 50%, diffuse positive > 50%, and negative (−) 0%. For the PD-L1 antibody, a total % of positive neoplastic cells and % of intervening stromal cells and lymphocytes was recorded.

### 2.1. Mutation Analysis

A mutation analysis detection of tumor and non-tumor tissue was performed using a TruSight Oncology 500 (TSO500) panel (Illumina, San Diego, CA, USA) [22]. In two cases, data from the TruSight Tumor 170 panel (TS170) (Illumina) were used for samples with low DNA quality. The gene list was previously published [23]. Total nucleic acid was extracted using an FFPE DNA kit (automated on an RSC 48 Instrument, Promega, Madison, WI, USA). Purified DNA was quantified using a Qubit Broad Range DNA assay (Thermo Fisher Scientific, Waltham, MA, USA). The quality of DNA was assessed using the FFPE QC kit (Illumina). DNA samples with Cq < 5 were used for further analysis. After DNA enzymatic fragmentation with a KAPA Frag Kit (Kapa Biosystems, Washington, MA, USA), DNA libraries were prepared with the TSO500/TS170 (Illumina) according to the manufacturer’s protocol. Sequencing was performed on the NextSeq 500 sequencer (Illumina) following the manufacturer’s recommendations. A data analysis was performed using the TSO500/TS170 application on the BaseSpace Sequence Hub (Illumina). DNA variant filtering and annotation were performed using the cloud-based tool Variant Interpreter (Illumina). A custom variant filter was set up including only variants with coding consequences at an allelic frequency of 5% and higher. The cut-off was set at 1% only in the case of mutations known in related tumor tissue. Comparing tumor and non-tumor data, germline alterations were excluded. The remaining subset of variants was checked visually, and suspected artefactual variants were excluded.

### 2.2. Analysis of VHL Promoter Methylation

The detection of promoter methylation was carried out via methylation-specific PCR as previously described [24].

### 2.3. LOH Analysis

For an LOH analysis of neoplastic tissue DNA, ten STR (short tandem repeats) markers D3S666, D3S1270, D3S1300, D3S1581, D3S1597, D3S1600, D3S1603, D3S1768, D3S2338 and D3S3630 located on the short arm of chromosome 3 (3p) were chosen from the database (Gene Bank UniSTS) [25].

## 3. Results

Four patients were enrolled in the study. Clinicopathologic data are summarized in Table 1. The patients were two men and two women, with ages ranging from 53.6 to 67.4 years (mean 61.5, median 62.5 years) at the time of renal surgery. Radical nephrectomy was performed in three cases. In one case, nephron sparing surgery was performed, but during the follow-up period, radical nephrectomy was completed due to recurrence (after a period of 72.6 months). Tumor size ranged from 30 mm to 75 mm (mean 53.5, median 54.5). The pathologic stage included 1× pT2a, 1× pT3a, and 2× pT1a. At the time of diagnosis, one patient had synchronous pulmonary lesions. The median progression-free interval (PFI) of the other cases was 40.5 months.

The mean age at the time of pulmonary metastasectomy was 65.5 years. Two patients had bilateral lung metastases, which were resected in a multistage fashion in independent surgeries. Overall, nine metastases were removed (in three patients, there were two metastases; in one patient, there were three metastases).

Signs of aggressive behavior were found approximately 2 to 35 months after pulmonary metastasectomy (metastatic progression to bones, lung, mediastinum, lymph nodes, and brain; median PFI was 18.7). Follow-up data were available for all patients, ranging approximately from 88 to 123 months (mean 104.4, median 103.4 months). For brain metastasis, surgical treatment using a gamma knife was performed. However, this patient died of peritonitis 3 months after the brain surgery. One patient died from the progression of the disease to the lung and bone 6 years after pulmonary surgery. To date, one patient with a progression of disease after 2 months (lymphatic tissue, bones, kidney) and one patient with a progression of disease 35 months (lymph nodes) after pulmonary surgery are alive.

### 3.1. Morphology

All cases showed morphologic features typical of CCRCC. Primary tumors were arranged in a solid alveolar pattern, and occasionally with smaller cystic areas. The rich vasculature characteristic of CCRCC was noted in all primary tumors. Only small foci of necrosis or regressive changes were recorded. Neoplastic cells were mostly voluminous with clear to pale eosinophilic cytoplasm. The histologic grade was 2 in three tumors and 3 in one tumor. Metastases showed relatively uniform morphology, arranged mostly in solid architecture and composed of predominantly clear cells. The histologic grade was 2 in 8/9 metastatic foci and 3 in 1/9 metastases (Table 2).

### 3.2. Immunohistochemical Analysis

All primary tumors and metastases were positive for CA9 (diffuse strong positivity) and vimentin. The Ki-67 proliferation index ranged from 3–12 positive cells/high-power field (under 10%). Primary tumors and metastases were negative for keratin 7.

The primary tumor and metastases were immunohistochemically examined using BAP1 antibody. Except for one tumor (patient 4), all primary tumors were BAP1 negative. In patient 3, negative BAP1 in the primary tumor and positive BAP1 in two of three PMs were documented.

PD-L1 reactivity was evaluated in all available samples. Only one primary tumor showed significant positivity (up to 30% of neoplastic cells); however, no positivity was documented in the available tissue from pulmonary metastasis (Table 3).

### 3.3. Molecular Genetic Analysis

Results of the molecular genetic analysis are summarized in Table 4. Typical *VHL* gene alterations were found in three primary tumors and their PMs (75%). In the patient without *VHL* mutation, we found alterations in *CUL3*, *DOT1L*, *SETD2* and *TSC1* in the primary tumor, with the addition of *BAP1* gene mutation in its analyzable PMs.

The comparison of mutation pattern among primary tumors and their PMs showed heterogeneity in three (75%) cases. In one case (patient 1), inter-metastatic differences were also found. In one metastasis, the mutation of *GNAQ* and loss of LOH3p were detected; however, in the second metastasis those changes were not confirmed. The comparison is displayed in Table 5.

## 4. Discussion

The loss of the short arm of chromosome 3 in CCRCC is a ubiquitous somatic event, accompanied by the inactivation of the remaining *VHL* gene through mutation or methylation (in >90%) [26,27,28,29].

The *VHL* gene product (pVHL) is a component of E3 ubiquitin ligase complex, a key regulator of the cellular response to hypoxia. The E3 ubiquitin ligase complex promotes the degradation of its substrates including the alpha subunit of the hypoxia inducible factor (HIFα). The loss of *VHL* results in the accumulation of HIF-α, leading to the constitutive expression of HIF target genes. These genes are involved in angiogenesis (e.g., *VEGF*), glycolysis and glucose transport (e.g., *GLUT1*), and erythropoiesis (e.g., *EPO*), which molecularly characterize CCRCC [30,31]. Mutations in other members of the E3 ubiquitin ligase complex such as elongin C (*ELOC/TCEB1*) and cullin 2 (*CUL2*) occur rarely and are mutually exclusive to *VHL*. Although there are differences between tumors with mutations in *TCEB1* and *VHL*, the dysregulation of the VHL pathway may explain the overlapping morphology and immunohistochemical profile [32].

Chromosome 3p loss may be identified using different molecular genetic methods. This and the mutation or promoter hypermethylation of *VHL* are so common in CCRCC that a subset of tumors without such alterations may be misclassified [33]; however, the usage of extensive molecular testing is rare in current clinical practice. Varying driver gene alterations underpin CCRCC evolution and biology [34,35]. CCRCCs with *VHL* loss as the only driver event are indolent and rarely metastasize.

The loss of 3p results in the simultaneous loss of three other tumor suppressor genes that are frequently mutated in CCRCC: *Polybromo 1* (*PBRM1*) (~50%), *SET domain containing 2* (*SETD2*) (~20%), and *BRCA1-associated protein 1* (*BAP1*) (~15%) [26,32,36]. It should be noted that tumorigenesis in CCRCC follows a trunk-branch evolution [37], in which the trunk mutation (*VHL*) is responsible for tumorigenesis and sub-clonal mutations (i.e., *PBRM1*, *SETD2*, *BAP1*) are developed during disease progression.

Similar to *VHL*, *PBRM1* is often mutated early during tumor development [38]. *PBRM1*-mutated tumors with subsequent *SETD2* mutations, driver somatic copy number alterations, or P13K pathway alterations have a more attenuated disease course [36,37,39]. In contrast, CCRCCs with *BAP1* mutations or multiple driver mutations are associated with aggressive clinical behavior and early metastatic disease. Additional driver mutations and somatic copy number alterations include (i) inactivating mutations in histone modifying genes (*KDM5C* and *KDM6A*), (ii) mutations in the mTOR pathway genes (*TSC1*, *TSC2*, *MTOR*, *PIK3CA*, *PTEN*), (iii) the loss of TP53, and (iv) losses of chromosomes 14 and 9 [26,34,40].

Recent large scale gene expression analyses of metastatic CCRCC identified unique molecular subsets with distinct drug response characteristics [38,41]. CCRCC with high angiogenic gene signatures had a favorable response to anti-angiogenic therapies and were enriched with PBRM1 loss [35,41]. In contrast, CCRCCs with an inflamed microenvironment were associated with the highest PD-L1 expression, preferential responsiveness to regimes containing immune checkpoint inhibitors and the highest rates of sarcomatoid change and *BAP1* mutations [38,39,41].

Passow et al. [11] also showed inter-metastatic tumor heterogeneity in BAP1 immunohistochemical reactivity in their study. The primary tumor in their study was BAP1 IHC positive, the first bone metastasis (synchronous) was IHC negative, and the second bone metastasis (diagnosed approximately 9 months later) was BAP1 IHC positive. In our study, we also observed variability in BAP1 immunohistochemical reactivity. In one of our cases (no. 3), the primary tumor and one of its metastases (PM3) were both BAP1 negative, whereas its two other distant metastases (PM1, PM2) were BAP1 positive. Of note, these IHC findings were consistent with the mutation analysis. BAP1 IHC expression also perfectly matched with the mutation profile in our fourth case, although two different *BAP1* mutations were unexpectedly found in the primary tumor and its PM. We assume that this phenomenon could be a result of genetic drift during tumor progression

There are two genetic “supergroups” in RCCs: the Krebs cycle group and the mTOR/TSC group. CCRCC is by far the most common example of the Krebs cycle group, whereas the mTOR/TSC group includes a number of newly recognized novel tumors such as eosinophilic solid and cystic RCC (ESC-RCC), eosinophilic vacuolated tumor (EVT), low-grade oncocytic tumor (LOT), and RCC with prominent fibromyomatous stroma (RCC FMS), for which the mutation of *TSC1*, *TSC2* and/or *M**TOR* is typical [42].

The mTOR pathway is an intracellular signaling pathway important for regulating the cell cycle. The most common genes involved in the tumorigenesis of the mTOR pathway group are *TSC1*, *TSC2*, and *MTOR.*

The mutation of the *TSC* genes in CCRCC is unusual but has been documented. Pang et al. [43] reported a rare case of CCRCC with novel biallelic somatic mutations in *TSC2*. This was a case of a 14-year-old female with VHL syndrome, where histologic findings were typical of CCRCC morphology. In addition, immunohistochemical findings also showed immunohistochemical expression for keratin, vimentin, CD10, and RCC, with negative results for CA9, keratin 7 and TFE3 staining. In our series, one of our patients (patient 1) demonstrated an interesting combination of mutations of *VHL* and *TSC1* in the PM, whereas we did not observe this phenomenon in the primary tumor. In the second patient, we verified a combination of *VHL* and *PTEN* mutations in the primary tumor and both metastases. In our third patient, the primary tumor showed a combination of *TSC1*, *CUL3*, *DOT1L* and *SETD2* gene mutations (but not *BAP1*), whereas the PM had the same genetic mutations plus *BAP1* mutation. This patient had metastatic disease at multiple sites post-surgery with disease progression. These molecular genetic findings indicate that in metastatic lesions, subclonal driver mutations are potentially responsible for spread and possible treatment failure. Such driver mutations were potentially missed due to sampling error or a lower number in samples analyzed by bulk sequencing. Another explanation might be the development of driver mutations over the course of the treatment. Current evidence suggests that treatment resistance and/or failure is caused by the resistant subclones, which were not targeted by the initial treatment [37]. We believe that optimizing the sampling approach in the metastatic setting, including the biopsy of newly developed metastatic CCRCC lesions, is important and can aid in effective therapeutic regimens due to the possible continued propagation of subclones.

One of the important novel renal entities in the differential diagnosis of CCRCC is RCC FMS [42]. Recognizing RCC FMS not only has academic value, but it also carries potential clinical implications and therapeutic management. Based on limited clinical data, these tumors tend to behave in an indolent fashion in most cases. In the largest cohort study of RCC FMS published to date [44], no evidence of recurrence or progression after surgical removal was documented. RCC FMS was included in the 2016 WHO classification of renal tumors as an emerging/provisional entity as “RCC with (angio) leiomyomatous stroma” [2]. However, distinct diagnostic criteria were not defined by the WHO classification. In the Genitourinary Pathology Society (GUPS) update review paper, the diagnostic histologic criteria for this distinct subtype of RCC have recently been established [42]. Tumors are composed of invariably voluminous epithelial clear cell components, which are typically diffusely positive for keratin 7 and of fibroleiomyomatous stroma. In this type of RCC, recurrent mutations involving the genes of the TSC/MTOR pathway were found. A subset of tumors with almost identical morphologic features showed mutations involving *ELOC* (also referred to as *TCEB1*), typically associated with the monosomy of chromosome 8 [44]. Both tumor subtypes lack *VHL* or chromosome 3p abnormalities [42,44]. In fact, it is not clear whether *TSC/MTOR* and *ELOC* mutated RCC with fibromyomatous stroma are two different tumor types, or just part of the molecular genetic variability within one tumor entity. Recently, one tumor with confirmed monosomy 8 and *ELOC* deletion as well as a *TSC1* mutation was documented [32,44].

RCC FMS are suggested to be more frequently sporadic; however, identical tumors were documented in patients with TSC. However, although the duration of the follow-up period is limited, most RCC FMS with *TSC/MTOR* mutations have demonstrated an indolent biological behavior [44]. However, lymph node metastases have been reported in rare cases associated with TSC recently. Although the initial report on *ELOC* (*TCEB1*)-associated RCC FMS suggested indolent behavior, an aggressive clinical course was recently described [45].

## 5. Conclusions

CCRCC are highly heterogeneous tumors, with complex molecular profiles both in the primary and metastatic settings. Tumor mutational profiles can be different not only between primary and metastatic tumors but also among multiple metastatic lesions themselves. It is evident that a one-size-fits-all approach is not optimal for treating advanced CCRCC and treatments need to be personalized. In this regard, optimizing tumor sampling and clinical management approaches in metastatic settings is crucial in order to identify subclonal mutations, which can ultimately lead to effective targeted therapies. The future of the successful personalized treatment and management of CCRCC is contingent upon a good understanding and accurate accounting for tumor heterogeneity.

The results of previously published studies and our own results show that CCRCC is a genetically heterogeneous tumor. The genetic background and mutation profile are highly variable within the primary tumor. However, data about the molecular genetic profile of the primary tumor and multiple metastases are very limited. It is apparent that the mutation profile can be different not only between the primary tumor and metastasis, but also among multiple metastases. Such important findings raise the question of the direct testing of each metastasis before the potential targeted therapy. Current clinical practice largely reflects genetic changes in primary tumors only. Because current oncologic treatment is reserved mostly for unresectable primary tumors and metastatic disease, we believe that such findings may become of critical importance.

## Figures and Tables

**Table 1 cancers-13-05906-t001:** Primary tumors: clinicopathological features.

	Patient 1	Patient 2	Patient 3	Patient 4
Sex	F	M	M	F
Age (years)	61.6	67.4	63.4	53.6
Size (mm)	39	75	30	70
pT (UICC 2017)	pT1a	pT2a	pT1a	pT3b
Grade (WHO/ISUP)	3	2	2	2
TTP meta 1	40.5	M1	59.6	38.1
TTP meta 2	40.5	M1	81.1	38.1
TTP meta 3	-	-	81.1	

F, female; M, male; M1, M1 stage (pulmonary metastases at the time of the renal cancer diagnosis); TTP, time to pulmonary progression (months).

**Table 2 cancers-13-05906-t002:** Grade of the primary tumors and metastases.

	Patient 1	Patient 2	Patient 3	Patient 4
Primary tumor grade	3	2	2	2
Met 1 grade	2	2	3	2
Met 2 grade	2	2	2	2
Met 3 grade			2	

Met, metastasis.

**Table 3 cancers-13-05906-t003:** PD-L1 reactivity in the primary tumor and metastases.

PD-L1	Case 1	Case 2	Case 3	Case 4
Primary tumor	* 0% ** 0%	* 0% ** 0%	* up to 5% ** 0%	* 30% ** 0%
Met 1	NA	* 0% ** 0%	* 0% ** up to 5%	* 0% ** 0%
Met 2	* up to 5% ** 0%	* 0% ** 0%	NA	NA
Met 3			NA	

Met, metastasis; * PD-L1 in neoplastic cells; ** PD-L1 in tumor infiltrating lymphocytes and stroma; NA, not available.

**Table 4 cancers-13-05906-t004:** Mutational profile of primary tumors and their PMs using TSO500/TS170 panels.

	Gene	Protein ID:Protein Alteration	Transcript ID: Mutation	Allele Frequency
Patient 1—primary tumor TMB—6.5	*MSH6*	NP_000170.1:p.(Ala780Ser)	NM_000179.2:c.2338G>T	0.2586
	*MYOD1*	NP_002469.2:p.(Glu158Lys)	NM_002478.4:c.472G>A	0.2071
	*PBRM1*	NP_060783.3:p.(Tyr893Ter)	NM_018313.4:c.2679T>A	0.2885
	*SETD2*	NP_054878.5:p.(Lys2471Ile)	NM_014159.6:c.7412A>T	0.3654
	* TFE3 *	NP_006512.2:p.(Pro374Ala)	NM_006521.5:c.1120C>G	0.2237
	*VHL*	NP_000542.1:p.(Ser65Ter)	NM_000551.3:c.194C>A	0.3855
Patient 1—metastasis 1 TMB—5.5	* BCORL1 *	NP_001171701.1:p.(Pro787Thr)	NM_001184772.2:c.2359C>A	0.2074
	*MSH6*	NP_000170.1:p.(Ala780Ser)	NM_000179.2:c.2338G>T	0.256
	*MYOD1*	NP_002469.2:p.(Glu158Lys)	NM_002478.4:c.472G>A	0.256
	*PBRM1*	NP_060783.3:p.(Tyr893Ter)	NM_018313.4:c.2679T>A	0.2657
	*SETD2*	NP_054878.5:p.(Lys2471Ile)	NM_014159.6:c.7412A>T	0.2372
	* TSC1 *	NP_000359.1:p.(Glu839Ter)	NM_000368.4:c.2515G>T	0.3039
	*VHL*	NP_000542.1:p.(Ser65Ter)	NM_000551.3:c.194C>A	0.2896
Patient 1—metastasis 2	* MSH6 *	NP_000170.1:p.(Ala780Ser)	NM_000179.2:c.2338G>T	0.30
	* TSC1 *	NP_000359.1:p.(Glu839Ter)	NM_000368.4:c.2515G>T	0.39
	* VHL *	NP_000542.1:p.(Ser65Ter)	NM_000551.3:c.194C>A	0.40
	* GNAQ *	NP_002063.2:p.(Tyr101Ter)	NM_002072.4:c.303C>A	0.08
Patient 2—primary tumor TMB—4.7	*CDK12*	NP_057591.2:p.(Leu529PhefsTer81)	NM_016507.2:c.1585del	0.1264
	*PTEN*		NM_000314.6:c.492+1del	0.2
	*REL*	NP_002899.1:p.(Ser274Cys)	NM_002908.3:c.821C>G	0.0685
	*SETD2*	NP_054878.5:p.(Gly1467ArgfsTer8)	NM_014159.6:c.4398dup	0.105
	*TGFBR2*	NP_001020018.1:p.(Glu510Asp)	NM_001024847.2:c.1530A>C	0.0857
	*VHL*		NM_000551.3:c.463+2T>A	0.1091
Patient 2—metastasis 1 TMB—6.3	*CDK12*	NP_057591.2:p.(Leu529PhefsTer81)	NM_016507.2:c.1585del	0.0993
	*PTEN*		NM_000314.6:c.492+1del	0.2207
	*REL*	NP_002899.1:p.(Ser274Cys)	NM_002908.3:c.821C>G	0.1138
	*SETD2*	NP_054878.5:p.(Gly1467ArgfsTer8)	NM_014159.6:c.4398dup	0.1229
	*TGFBR2*	NP_001020018.1:p.(Glu510Asp)	NM_001024847.2:c.1530A>C	0.15
	*VHL*		NM_000551.3:c.463+2T>A	0.1044
Patient 2—metastasis 2 TMB—7.1	*CDK12*	NP_057591.2:p.(Leu529PhefsTer81)	NM_016507.2:c.1585del	0.0989
	*PTEN*		NM_000314.6:c.492+1del	0.2308
	*REL*	NP_002899.1:p.(Ser274Cys)	NM_002908.3:c.821C>G	0.1059
	*SETD2*	NP_054878.5:p.(Gly1467ArgfsTer8)	NM_014159.6:c.4398dup	0.1365
	*TGFBR2*	NP_001020018.1:p.(Glu510Asp)	NM_001024847.2:c.1530A>C	0.0856
	*VHL*		NM_000551.3:c.463+2T>A	0.0828
Patient 3—primary tumor TMB—4.7	*CUL3*	NP_001244127.1:p.(Val452PhefsTer9)	NM_001257198.1:c.1354del	0.1523
	*DOT1L*	NP_115871.1:p.(Met147Ile)	NM_032482.2:c.441G>T	0.1664
	*SETD2*	NP_054878.5:p.(Gln2070Ter)	NM_014159.6:c.6208C>T	0.1696
	*TSC1*	NP_000359.1:p.(Asn364LysfsTer5)	NM_000368.4:c.1091dup	0.195
Patient 3—metastasis 1 TMB—4.7	* BAP1 *	NP_004647.1:p.(Arg385Ter)	NM_004656.3:c.1153C>T	0.1193
	*CUL3*	NP_001244127.1:p.(Val452PhefsTer9)	NM_001257198.1:c.1354del	0.1127
	*DOT1L*	NP_115871.1:p.(Met147Ile)	NM_032482.2:c.441G>T	0.1032
	*SETD2*	NP_054878.5:p.(Gln2070Ter)	NM_014159.6:c.6208C>T	0.1191
	*TSC1*	NP_000359.1:p.(Asn364LysfsTer5)	NM_000368.4:c.1091dup	0.0714
Patient 3—metastasis 2 TMB—4	* BAP1 *	NP_004647.1:p.(Arg385Ter)	NM_004656.3:c.1153C>T	0.1026
	*CUL3*	NP_001244127.1:p.(Val452PhefsTer9)	NM_001257198.1:c.1354del	0.1076
	*DOT1L*	NP_115871.1:p.(Met147Ile)	NM_032482.2:c.441G>T	0.0642
	*SETD2*	NP_054878.5:p.(Gln2070Ter)	NM_014159.6:c.6208C>T	0.1177
	*TSC1*	NP_000359.1:p.(Asn364LysfsTer5)	NM_000368.4:c.1091dup	0.0559
Patient 4—primary tumor TMB—3.1	*ARID5B*	NP_115575.1:p.(Ala954Asp)	NM_032199.2:c.2861C>A	0.0835
	* BAP1 *	NP_004647.1:p.(Gly703SerfsTer30)	NM_004656.3:c.2107_2116del	0.1046
	*VHL*	NP_000542.1:p.(Arg69AlafsTer82)	NM_000551.3:c.201_225del	0.0917
	* XIAP *	NP_001158.2:p.(Ser169Tyr)	NM_001167.3:c.506C>A	0.0583
Patient 4—metastasis 1 TMB—1.6	*ARID5B*	NP_115575.1:p.(Ala954Asp)	NM_032199.2:c.2861C>A	0.0835
	* BAP1 *		NM_004656.3:c.122+1G>T	0.0806
	*VHL*	NP_000542.1:p.(Arg69AlafsTer82)	NM_000551.3:c.201_225del	0.0718
Patient 4—metastasis 2	* BAP1 *		NM_004656.3:c.122+1G>T	0.03
	* VHL *	NP_000542.1:p.(Arg69AlafsTer82)	NM_000551.3:c.201_225del	0.06

Tumor/metastasis differences highlighted by red color. TMB, Tumor Mutation Burden; TS170 data in gray.

**Table 5 cancers-13-05906-t005:** The genetic profile of primary tumors and their PMs.

	Patient 1 Primary Tumor	Patient 1 Metastasis 1	Patient 1 Metastasis 2	Patient 2 Primary Tumor	Patient 2 Metastasis 1	Patient 2 Metastasis 2	Patient 3 Primary Tumor	Patient 3 Metastasis 1	Patient 3 Metastasis 2	Patient 3 Metastasis 3	Patient 4 Primary Tumor	Patient 4 Metastasis 1	Patient 4 Metastasis 2
*ARID5B*													
*BAP1*													
*BCORL1*													
*CDK12*													
*CUL3*													
*DOT1L*													
*GNAQ*													
*MSH6*													
*MYOD1*													
*PBRM1*													
*PTEN*													
*REL*													
*SETD2*													
*TFE3*													
*TGFBR2*													
*TSC1*													
*VHL*													
*XIAP*													
*VHL LOH*													
*VHL M*													
		non-sense mutation											
		missense mutation											
		splice-site mutation											
		not analyzable											
		*LOH*—loss of heterozygosity—positive											
		*LOH*—loss of heterozygosity—negative											
		*LOH*—loss of heterozygosity—borderline											
		M—promoter methylation—positive											
		M—promoter methylation—negative											

## Data Availability

Data is contained within the article.

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
