# Peer review of "Mutation Profile Variability in the Primary Tumor and Multiple Pulmonary Metastases of Clear Cell Renal Cell Carcinoma. A Review of the Literature and Analysis of Four Metastatic Cases"

_cancers, 2021, doi:10.3390/cancers13235906_

Round 1
Reviewer 1 Report
The authors present a review on the published data regarding intra- and inter-tumoral mutation profile variability in primary and pulmonary metastasis of clear cell renal carcinoma (CCRCC) combined with analysis of data collected from set of four patient. The authors discuss the commonly genetic alterations contributing to CCRCC and their observed variability intra-tumorally and/or between primary and metastatic tumors using available research literature. Additionally, the authors include their own study on mutation profiles and the changes in these profiles among primary and metastatic tumors using data obtained from four selected patients. The article provides an overview on the genetic features of CCRCC but also highlights the importance of more detailed analysis of tumor heterogeneity to identify treatment resistance driving gene mutations in cancer cell subclones to allow improved cancer therapy outcomes. Few minor revisions are suggested.
- In the last sentence of the simple summary, authors use the term “in our series of CCRCC”. This presumably refers to the set of four patients used in the analysis, but the authors should clarify this to avoid confusion. Same applies to sentence in line 32.
- In line 57, there should be “on chromosome 3p” instead of “on 3p”.
- In line 57 the authors mention about the chromosome 3p loss not being entirely specific for CCRCC but do not elaborate the matter further leaving the reader wonder why it was necessary to mention. Authors could either explain very briefly the relevance of chromosome 3p deletion in the context of other cancers or leave the notion about it not being specific to CCRCC out.
- In row 233, “2.3 to 35.1 months” should be replaced with “approximately 2 to 35 months” as in this particular case expression of time period in decimal accuracy does not make sense. Same applies for the time period mentioned at the end of line 235/beginning of line 236.
- In line 240-241, there is an extra “after”. There should be “…progression of disease 35 months after pulmonary surgery…” instead of “…progression of disease after 35 months after pulmonary surgery..”.
- Minor typographical error in line 340: should read “TSC1” but instead there is “TCS1”. Should be corrected.
- In line 368 there is a minor typographical error, should be “FMS” but there is “MFS” instead.
Author Response
Response to Reviewer 1:
First of all, thank you for positive evaluation of our work. We followed all points from review as follows.
Point 1: In the last sentence of the simple summary, authors use the term “in our series of CCRCC”. This presumably refers to the set of four patients used in the analysis, but the authors should clarify this to avoid confusion. The same applies to sentence in line 32.
Response 1: both sentences have been modified: “cohort of 4 patients with metastatic CCRCC“.
Point 2: In line 57, there should be “on chromosome 3p” instead of “on 3p”.
Response 2: corrected according to the recommendation.
Point 3: In line 57 the authors mention about the chromosome 3p loss not being entirely specific for CCRCC but do not elaborate the matter further leaving the reader wonder why it was necessary to mention. Authors could either explain very briefly the relevance of chromosome 3p deletion in the context of other cancers or leave the notion about it not being specific to CCRCC out.
Response 3: Authors would like to point out the fact, that in difficult diagnostic pathology cases, molecular evaluation can be used to support a diagnosis of CCRCC, such as chromosome 3p loss (FISH, cytogenetics, or copy number analysis) or VHL mutational analysis, with the understanding that 3p loss may not be entirely specific for clear cell RCC in all contexts.
References: (Williamson, S.R.; Halat, S.; Eble, J.N.; Grignon, D.J.; Lopez-Beltran, A.; Montironi, R.; Tan, P.-H.; Wang, M.; Zhang, S.; MacLennan, G.T.; et al. Multilocular Cystic Renal Cell Carcinoma: Similarities and Differences in Immunoprofile Compared With Clear Cell Renal Cell Carcinoma. The American Journal of Surgical Pathology 2012, 36).
Correction: Chromosome arm 3p loss is a common event in primary CCRCC, in difficult diagnostic pathology cases, molecular evaluation can be used to support a diagnosis of CCRCC, such as chromosome 3p loss (FISH, cytogenetics, or copy number analysis) or VHL mutational analysis. However, 3p loss may not be entirely specific for clear cell RCC in all contexts [15]. For example, chromosome 3p loss has been recognized in subsets of papillary RCC, unclassified RCC, and RCC with amplification of the 6p21 / TFEB gene region, including in tumors with non-clear cell morphology and without VHL alterations.
Point 4: In row 233, “2.3 to 35.1 months” should be replaced with “approximately 2 to 35 months” as in this particular case expression of time period in decimal accuracy does not make sense. Same applies for the time period mentioned at the end of line 235/beginning of line 236.
Response 4: corrected according to the reviewer´s recommendation.
Point 5: In line 240-241, there is an extra “after”. There should be “…progression of disease 35 months after pulmonary surgery…” instead of “…progression of disease after 35 months after pulmonary surgery..”.
Response 5: corrected according to the reviewer´s recommendation.
Point 6: Minor typographical error in line 340: should read “TSC1” but instead there is “TCS1”. Should be corrected.
Response 6: corrected according to the reviewer´s recommendation.
Point 7: In line 368 there is a minor typographical error, should be “FMS” but there is “MFS” instead.
Response 7: corrected according to reviewer´s recommendation.
-------------------------------------------------------------------------------------------------------------------------------
Other changes: The highlighted sections have been reduced as requested by reviewers/editor.
- Introduction – line 47, 48 – Clear cell renal cell carcinoma (CCRCC) is the most common renal carcinoma, accounting for more than 70% of all malignant renal neoplasms in adult population [1,2]. Correction: Clear cell renal cell carcinoma (CCRCC) is the most common renal carcinoma, accounting for more than 70% of adult renal cancer.
- line 69 – 74 - The comprehensive molecular characterization of CCRCC carried out by the Cancer Genome Atlas Research Network through a whole exome sequencing of more than 400 CCRCC samples identified 19 significantly mutated genes, including VHL, PBRM1, SETD2, KDMC, PTEN, BAP1, MTOR and TP53, representing the eight most frequently mutated genes.
Correction: The molecular study of the Cancer Genome Atlas Research Network identified 19 significantly mutated genes, with alterations of VHL, PBRM1, SETD2, KDMC, PTEN, BAP1, MTOR and TP53, being the eight most frequent.
- line 75 – 77 – CCRCC is ideal for studying intra-tumoral heterogeneity, since current guidelines do not recommended adjuvant therapy after surgery.
Correction: CCRCC is ideal for studying intra-tumoral heterogeneity, since adjuvant therapy is not standard practice.
1.1.2. Inter-tumoral heterogeneity: line 104 – 106 – Eckel-Passow and colleagues analyzed BAP1 and PBRM1 loss of protein expression, using immunohistochemical assay, in patient-matched primary and metastatic tumors from 97 patients. In their cohort, 20% were loss of BAP1 and 57% showed loss of PBRM1 expression in their primary tumors.
Correction: Eckel-Passow and colleagues analyzed BAP1 and PBRM1 loss of protein expression, using an immunohistochemical assay, in patient-matched primary and metastatic tumors from 97 patients. In their cohort, 20% exhibited loss of BAP1 and 57% showed loss of PBRM1 in the primary tumors.
1.1.3. Inter-metastatic heterogeneity: line 115 – 117 – The primary tumor for this patient was BAP1 positive, while the first bone metastasis (diagnosed approximately 9 months later) was IHC negative, the second bone metastasis) was IHC positive.
Correction: The primary tumor for this patient was BAP1 positive, whereas the first bone metastasis was IHC negative and the second bone metastasis) was IHC positive.
Material and Methods – line 170 – 174 – this part has been removed.
Replaced by: The tissue were processed as published previously [21].
Tables 1-4 have been completely removed – replaced by references – Line 189 - Mutation analysis detection of tumor and non-tumor tissue was performed using TruSight Oncology 500 (TSO500) panel (Illumina, San Diego, CA) - [22]. Line 190, 191 - The gene list was previously published [23]. Line 211-220 - Detection of promoter methylation was carried out via methylation specific PCR as previously described [24]. Line 223 – 233 - For LOH analysis of neoplastic tissue DNA, ten STR (short tandem repeats) markers D3S666, D3S1270, D3S1300, D3S1581, D3S1597, D3S1600, D3S1603, D3S1768, D3S2338 and D3S3630 located on the short arm of chromosome 3 (3p) were chosen from the database (Gene Bank UniSTS) [25].
Other minor changes:
Introduction: line 80 – the morphologic (instead of “morphologic”); levels (instead of level).
1.1.1. Intra-tumoral heterogeneity: line 90 - that (instead of „which“)
1.1.2. Inter-tumoral heterogeneity: line 107 – sections (instead of “section”)
1.1.3. Inter-metastatic heterogeneity: line 115 – patient (instead of “case”); line 120 – comma inserted - The limitation of this study was that the expression was determined using an immunohistochemical assay only, with no further molecular genetic validation.
1.2.1. Intra-tumoral heterogeneity: line 129 – resulting in (instead of “converting on”); line 133 – multiregional (instead of “multiregion”)
1.2.2. Inter-tumoral heterogeneity: line 144 – into (instead of “to”); subtypes (instead of “subtype”)
1.2.3. Inter-metastatic heterogeneity: line 149 – Thirty (instead of “30”)
- Materials and Methods: line 177 – added - PD-L1 (22C3, monoclonal, Cell Signaling, Danvers, MA, 1:25); line 184 - For PD-L1 antibody, total % of positive neoplastic cells and % of intervening stromal cells and lymphocytes was recorded.
2.1. Mutation Analysis: line 203 – mutations known (instead of “mutation profile known”); the cut-off (instead “cut-off”)
- Results: line 246 – metastases (instead of “metastases”); line 247 - patients (instead of “cases”); line 248 – patient (instead of “case”)
3.1. Morphology: line 264 – a solid (instead of “solid”; line 269- predominantly (instead of “mainly”)
3.2. Immunohistochemical Analysis: line 274 and 276 – and (instead of “ as well as”);line 279 – tumor (instead of “case”). Line 282 – 284 – added: PD-L1 reactivity was evaluated in all available samples. Only 1 primary tumor showed significant positivity (up to 30% of neoplastic cells), however no positivity was documented in available tissue from pulmonary metastasis (Table 3).
This new Table 3 has been added – PD-L1 reactivity in the primary tumor and metastases.
3.3. Molecular genetic analysis: line 292 – alterations were (instead “alteration was”); line 293 – without VHL mutation (instead of “without typical mutation”); we found alterations (instead of “alterations”)
Table 4. Mutational profile of primary tumors and its PMs using TSO500/TS170 panels: Tumor/metastasis differences highlighted by red colour. Results of TMB have been added.
- Discussion: paragraph 2, line 320 – 324 – However, absence of these alterations is rare and such RCCs may be misclassified [30]. On the other hand, mutation, promoter hypermethylation, and copy number analysis in RCCs are rarely utilized in daily routine practice.
Correction: This and mutation or promoter hypermethylation of VHL are so common in CCRCC that a subset of tumors without such alterations may be misclassified [33]; however, usage of extensive molecular testing is rare in current clinical practice.
paragraph 4, line 337 – 338 – In addition, these molecular changes may influence metastatic organ tropism [17] - the sentence has been removed
Author Contributions: editing authors' initials

Reviewer 2 Report
Re: review
First, I would like to congratulate the authors for their very well conducted and reported study. This manuscript describes the tumor heterogeneity in a pool of 4 patients with metastatic RCC. This is a very interesting study with clinical implications as it includes a special category of patients with cancer. The hypothesis is clearly formulated and the results are well presented and concise. Despite few inaccuracies in using a proficient scientific English, the manuscript overall is written well and the results are presented in a clear and logic format.
The manuscript can be improved by adding PD-L1 expression in the metastases vs. primary tumor, number of single nucleotide polymorphism, and tumor neoantigen burden.
As a minor point, some critical references are missing.
Author Response
Response to Reviewer 2:
Thank you again for positive evaluation of our work. We tried to follow all points from your review, however we were very limited by material availability.
Point 1: The manuscript can be improved by adding PD-L1 expression in the metastases vs. primary tumor:
Response 1: clone 22C3 was used and slides were evaluated accordingly. Section was added into the material and methods and in results. Unfortunately, we do not have enough of tissue from metastatic foci and we are not able to compare all neoplastic samples (Table 3).
Point 2: tumor neoantigen burden.
Response 2: Tumor Mutation Burden has been added in all samples, where results are available (Tab. 4).
Point 3: number of single nucleotide polymorphism
Response 3: We are not sure, what reviewer suggested as “number of single nucleotide polymorphism“. Should we add number of germ SNP in samples?
---------------------------------------------------------------------------------------------------------------------------------
Other changes: The highlighted sections have been reduced as requested by reviewers/editor.
- Introduction – line 47, 48 – Clear cell renal cell carcinoma (CCRCC) is the most common renal carcinoma, accounting for more than 70% of all malignant renal neoplasms in adult population [1,2]. Correction: Clear cell renal cell carcinoma (CCRCC) is the most common renal carcinoma, accounting for more than 70% of adult renal cancer.
- line 69 – 74 - The comprehensive molecular characterization of CCRCC carried out by the Cancer Genome Atlas Research Network through a whole exome sequencing of more than 400 CCRCC samples identified 19 significantly mutated genes, including VHL, PBRM1, SETD2, KDMC, PTEN, BAP1, MTOR and TP53, representing the eight most frequently mutated genes.
Correction: The molecular study of the Cancer Genome Atlas Research Network identified 19 significantly mutated genes, with alterations of VHL, PBRM1, SETD2, KDMC, PTEN, BAP1, MTOR and TP53, being the eight most frequent.
- line 75 – 77 – CCRCC is ideal for studying intra-tumoral heterogeneity, since current guidelines do not recommended adjuvant therapy after surgery.
Correction: CCRCC is ideal for studying intra-tumoral heterogeneity, since adjuvant therapy is not standard practice.
1.1.2. Inter-tumoral heterogeneity: line 104 – 106 – Eckel-Passow and colleagues analyzed BAP1 and PBRM1 loss of protein expression, using immunohistochemical assay, in patient-matched primary and metastatic tumors from 97 patients. In their cohort, 20% were loss of BAP1 and 57% showed loss of PBRM1 expression in their primary tumors.
Correction: Eckel-Passow and colleagues analyzed BAP1 and PBRM1 loss of protein expression, using an immunohistochemical assay, in patient-matched primary and metastatic tumors from 97 patients. In their cohort, 20% exhibited loss of BAP1 and 57% showed loss of PBRM1 in the primary tumors.
1.1.3. Inter-metastatic heterogeneity: line 115 – 117 – The primary tumor for this patient was BAP1 positive, while the first bone metastasis (diagnosed approximately 9 months later) was IHC negative, the second bone metastasis) was IHC positive.
Correction: The primary tumor for this patient was BAP1 positive, whereas the first bone metastasis was IHC negative and the second bone metastasis) was IHC positive.
Material and Methods – line 170 – 174 – this part has been removed.
Replaced by: The tissue were processed as published previously [21].
Tables 1-4 have been completely removed – replaced by references – Line 189 - Mutation analysis detection of tumor and non-tumor tissue was performed using TruSight Oncology 500 (TSO500) panel (Illumina, San Diego, CA) - [22]. Line 190, 191 - The gene list was previously published [23]. Line 211-220 - Detection of promoter methylation was carried out via methylation specific PCR as previously described [24]. Line 223 – 233 - For LOH analysis of neoplastic tissue DNA, ten STR (short tandem repeats) markers D3S666, D3S1270, D3S1300, D3S1581, D3S1597, D3S1600, D3S1603, D3S1768, D3S2338 and D3S3630 located on the short arm of chromosome 3 (3p) were chosen from the database (Gene Bank UniSTS) [25].
Other minor changes:
Introduction: line 80 – the morphologic (instead of “morphologic”); levels (instead of level).
1.1.1. Intra-tumoral heterogeneity: line 90 - that (instead of „which“)
1.1.2. Inter-tumoral heterogeneity: line 107 – sections (instead of “section”)
1.1.3. Inter-metastatic heterogeneity: line 115 – patient (instead of “case”); line 120 – comma inserted - The limitation of this study was that the expression was determined using an immunohistochemical assay only, with no further molecular genetic validation.
1.2.1. Intra-tumoral heterogeneity: line 129 – resulting in (instead of “converting on”); line 133 – multiregional (instead of “multiregion”)
1.2.2. Inter-tumoral heterogeneity: line 144 – into (instead of “to”); subtypes (instead of “subtype”)
1.2.3. Inter-metastatic heterogeneity: line 149 – Thirty (instead of “30”)
- Materials and Methods: line 177 – added - PD-L1 (22C3, monoclonal, Cell Signaling, Danvers, MA, 1:25); line 184 - For PD-L1 antibody, total % of positive neoplastic cells and % of intervening stromal cells and lymphocytes was recorded.
2.1. Mutation Analysis: line 203 – mutations known (instead of “mutation profile known”); the cut-off (instead “cut-off”)
- Results: line 246 – metastases (instead of “metastases”); line 247 - patients (instead of “cases”); line 248 – patient (instead of “case”)
3.1. Morphology: line 264 – a solid (instead of “solid”; line 269- predominantly (instead of “mainly”)
3.2. Immunohistochemical Analysis: line 274 and 276 – and (instead of “ as well as”);line 279 – tumor (instead of “case”). Line 282 – 284 – added: PD-L1 reactivity was evaluated in all available samples.
Only 1 primary tumor showed significant positivity (up to 30% of neoplastic cells), however no positivity was documented in available tissue from pulmonary metastasis (Table 3).
This new Table 3 has been added – PD-L1 reactivity in the primary tumor and metastases.
3.3. Molecular genetic analysis: line 292 – alterations were (instead “alteration was”); line 293 – without VHL mutation (instead of “without typical mutation”); we found alterations (instead of “alterations”)
Table 4. Mutational profile of primary tumors and its PMs using TSO500/TS170 panels: Tumor/metastasis differences highlighted by red colour. Results of TMB have been added.
- Discussion: paragraph 2, line 320 – 324 – However, absence of these alterations is rare and such RCCs may be misclassified [30]. On the other hand, mutation, promoter hypermethylation, and copy number analysis in RCCs are rarely utilized in daily routine practice.
Correction: This and mutation or promoter hypermethylation of VHL are so common in CCRCC that a subset of tumors without such alterations may be misclassified [33]; however, usage of extensive molecular testing is rare in current clinical practice.
paragraph 4, line 337 – 338 – In addition, these molecular changes may influence metastatic organ tropism [17] - the sentence has been removed
Author Contributions: editing authors' initials
